# A Novel Early Cretaceous Flower and Its Implications on Flower Derivation

**DOI:** 10.3390/biology11071036

**Published:** 2022-07-11

**Authors:** Xin Wang

**Affiliations:** State Key Laboratory of Palaeobiology and Stratigraphy, CAS Center for Excellence in Life and Paleoenvironment, Nanjing Institute of Geology and Palaeontology, Chinese Academy of Sciences, Nanjing 210008, China; xinwang@nigpas.ac.cn

**Keywords:** angiosperm, fossil, flower, evolution, Cretaceous

## Abstract

**Simple Summary:**

Many people are fascinated by beautiful flowers. Botanists have been intensively studying flowers to decipher their history. Various hypotheses have been advanced to account for the evolution of flowers. Unfortunately, most hypotheses are based on the understanding of modern flowers. Magnolia and related plants were once thought to bear the greatest resemblance to ancestral flowers. Naturally, botanists have been frequently surprised by new finds in the past century. A new flower named *Lingyuananthus* from the Early Cretaceous (about 125 million years ago) in China is the latest surprise for botanists. Distinct from all previously reported fossil flowers of similar ages, *Lingyuananthus* has an inferior ovary, syncarpy, and hypanthium. All three features were thought to be highly derived in angiosperms. Considering its Early Cretaceous age (the currently widely accepted earliest age of angiosperms and flowers), *Lingyuananthus* is a black swan that illustrates how astray botanists may have gone previously. Unable to account for the fossil flower diversity in the Early Cretaceous, botanical theories face a crisis. Among existing hypotheses, Zimmermann’s hypothesis, a long-time unfavored hypothesis, stands out like a black swan: it provides a better explanation for the origin and evolution of *Lingyuananthus*.

**Abstract:**

Background: The origin and early evolution of angiosperms, by far the most important plant group for human beings, are questions demanding answers, mainly due to a lack of related fossils. The Yixian Formation (Lower Cretaceous) is famous for its fossils of early angiosperms, and several Early Cretaceous angiosperms with apocarpous gynoecia have been documented. However, a hypanthium and an inferior ovary are lacking in these fossil angiosperms. Methods: The specimen was collected from the outcrop of the Yixian Formation in Dawangzhangzi in the suburb of Lingyuan, Liaoning, China. The specimen was photographed using a Nikon D200 digital camera, and its details were photographed using a Nikon SMZ1500 stereomicroscope and a MAIA3 TESCAN SEM. Results: A fossil angiosperm, *Lingyuananthus inexpectus* gen. et sp. nov, is reported from the Lower Cretaceous of China. Differing from those documented previously, *Lingyuananthus* has a hypanthium, an inferior ovary, and ovules inside its ovary. Such a character assemblage indicates its angiospermous affinity, although not expected by any existing leading angiosperm evolutionary theory. Conclusions: New fossil material with a unique character assemblage falls beyond the expectation of the currently widely accepted theories of angiosperm evolution. Together with independently documented fossils of early angiosperms, *Lingyuananthus* suggests that at least some early angiosperms’ flowers can be derived in a way that has been ignored previously.

## 1. Introduction

The Early Cretaceous is currently the widely accepted earliest age for angiosperms, and the Yixian Formation, which is famous for its abundant fossil angiosperms [1,2,3,4,5,6,7,8,9,10,11,12,13,14,15,16], belongs to the Lower Cretaceous (Barremian to Aptian, approximately 125 Ma) [17]. Theoretically, conduplicate carpels and their derivatives were thought to be ancestral among angiosperms [18]. This concept used to be widely accepted and frequently taught in classrooms. Although supported by some early angiosperms [2,3,4,5,8,10], this hypothesis is now facing increasing challenges from fossil evidence [1,7,11,14,19,20,21,22,23] and molecular studies [24,25,26,27,28,29]. Furthermore, a strong challenge is from the early Jurassic *Nanjinganthus* [21,22,30], which shows syncarpy and an inferior ovary, both unexpected for the assumed basalmost angiosperms [18]. Actually, *Nanjinganthus* is not peerless in challenging the traditional theories. *Monetianthus* Friis et al. 2009, a fossil representative of Nymphaeaceae [31], also has an inferior ovary and syncarpy. Other than *Nanjinganthus* and *Monetianthus*, fossil evidence of early angiosperms has raised similar challenges [1,6,9,11,12,14,15,20,32]. Here, I report a new angiosperm, *Lingyuananthus inexpectus* gen. et sp. nov, from the Yixian Formation (Lower Cretaceous, Barremian–Aptian) of Lingyuan, Liaoning, China. Distinct from early angiosperms previously reported from the formation, *Lingyuananthus* is a flower with an inferior ovary and multiple linear tepals arranged on the edge of the hypanthium. This not only adds to the great diversity of angiosperms in the formation but also joins *Nanjinganthus* and *Monetianthus* in demanding attention to a novel way to derive flowers in angiosperms.

## 2. Materials and Methods

The specimen was collected from the Yixian Formation near Dawangzhangzi Village, Lingyuan, Liaoning, China (41°15′ N, 119°15′ E) (Figure 1a–c). The fossiliferous layer is a siltstone of the Daxinfangzi layer in the formation (Figure 1c). The specimen includes two facing parts preserved as a compression with some coalified residue (Figure 2a). The locality has yielded *Archaefructus sinensis* Sun et al. 2002 [3]*, Sinocarpus decussatus* Leng and Friis 2003 [4,5], *Nothodichocarpum lingyuanensis* Han et al. 2017 [10]*, Neofructus lingyuanensis* Liu and Wang 2018 [8], and *Callianthus dilae* Wang and Zheng 2009 [32] (Table 1). The specimen is preserved on gray siltstone slabs approximately 11 cm × 11 cm. The specimen was photographed using a Nikon D200 digital camera and a Nikon SMZ1500 stereomicroscope with a Nikon DS-Fi1 digital camera. SEM was performed using a MAIA3 TESCAN housed at the Nanjing Institute of Geology and Palaeontology, Nanjing, China. All figures were organized using Photoshop 7.0.

## 3. Results


**Genus *Lingyuananthus* gen. nov**


**Diagnosis**: Flower syncarpous, with an elongated pedicel, ovary, and tepals. Ovary cup-formed, flat-topped, surrounded by a hypanthium with asymmetrically arranged lateral appendages. Multiple tepals, linear, on hypanthium edge or surface. Multiple ovules/seeds within the ovary.


**Species *Lingyuananthus inexpectus* gen. et sp. nov**


**Diagnosis**: The same as the genus.

**Description**: The fossil is 6.4 cm long and 3.8 cm wide, including a pedicel, an ovary, and multiple tepals (Figure 2a). The pedicel is slightly curved, 13 mm long, and 1.5 mm wide (Figure 2a). The fossil is syncarpous, with an ovary that is cup-shaped, 13.5 mm long, widest at the top, and 10 mm wide (Figure 2b, Figure 3a, and Figure 4d). The outer surface of the hypanthium has multiple lateral appendages, which tend to be concentrated on one side of the hypanthium (Figure 2a,b and Figure 4d). The floral roof is generally flat, secluding the ovary, with a small process (Figure 2b and Figure 4d,e). The tepals are linear, 0.7 mm wide, and 41 mm long, and not differentiated in size and morphology (Figure 2a, Figure 3d, and Figure 5). At least 15 tepals are arranged along the hypanthium edge and outer surface (Figure 2a,b and Figure 5). Multiple (more than three) ovules of variable forms are in the ovary, 0.48–0.92 mm wide and 1.16–1.3 mm long (Figure 2b–e, Figure 3b,c, Figure 4a, and Figure 5). Round papillae are seen on the ovule surface, 16–23 μm in diameter (Figure 4b). Transmitting tissue is seen under the floral roof (Figure 4c). No trace of a seed coat is seen throughout the ovary. 

**Remarks**: The disadvantage of the present study is its single specimen, undermining the validity of the conclusion. However, this does not reduce the significance of the fossil, as it follows the routine practiced in paleobotany. For example, both *Archaefructus* [2] and *Monetianthus* [33], published in *Science* and *Nature*, respectively, are based on only one specimen.

Due to its compression preservation, it is hard to judge the symmetry of the flower. However, the asymmetrical arrangement of lateral appendages on the outer surface of the ovary wall (hypanthium) seems to suggest that the flower of *Lingyuananthus* more likely has bilateral symmetry.

No trace of stamens or pollen grains is seen in the specimen, even when examined using SEM. This leads to the conclusion that *Lingyuananthus* is an incomplete flower, namely, a pistillate flower.

The “tepals” of *Lingyuananthus* are uniform in morphology: linear, with smooth margins, attached to the hypanthium edge or outer surface. They constitute the “perianth” in *Lingyuananthus*. Their uniform morphology suggests that they should not be termed “sepals” or “petals” (differentiated perianth elements) and are better termed as “tepals” (undifferentiated perianth elements). The preservation of the present specimen does not allow the author to tell the exact arrangement of the tepals (whorled or not, number of whorls) in *Lingyuananthus*.

The number of tepals is characterized as “multiple” since there are more than 3 (approximately 15) tepals preserved in the specimen. The exact number of tepals is to be determined by future studies of more specimens. The venation in the tepals is most likely parallel, but this cannot be ascertained with the current specimen.

Theoretically, ovules within a single ovary should be of uniform shape, but the outlines of ovules in *Lingyuananthus* vary. This variation may be due to different orientations, developmental stages, and preservation of the ovules.

Since *Lingyuananthus* is an obvious reproductive organ, the dimension and lack of pollen traces in the fossil leaves only one alternative, that it is a female reproductive organ. The inner bodies are millimetric in size, so they are interpreted as ovules because a seed coat, expected for seeds, is not seen in the fossil. These ovules are embedded within a closed structure, suggesting angio-ovuly in *Lingyuananthus*. This leads me to interpret the fossil as a flower rather than a fruit.

**Horizon**: The Yixian Formation, Barremian–Aptian, Lower Cretaceous.

**Holotype**: PB328297.

**Etymology**: *Lingyuan*- for the fossil locality, Lingyuan City; -*anthus* for flower; *inexpectus* for unexpected morphology of the flower.

**Depository**: The Herbarium, Nanjing Institute of Geology and Palaeontology, Nanjing, China.

## 4. Discussion

Angiosperms are the most diversified plant group on Earth. There are more than 300,000 species of angiosperms, accounting for more than 90% of the species diversity of land plants [34]. Ecologically and historically, angiosperms are important elements in the background for the origin and evolution of human beings. Considering the importance of angiosperms, it is rather natural that the origin and evolution of angiosperms have been constant foci of botanical studies for a long time. Literally, angiosperms are distinguished from their gymnosperm peers from their defining feature, an enclosed seed. However, all organisms tend to protect their offspring in one way or another [35]; e.g., some gymnosperms (e.g., conifers) also protect their seeds by secluding them after pollination [36]. Therefore, distinguishing angiosperms and gymnosperms using angiospermy alone is not completely accurate. A more critical survey indicates that a constant difference between angiosperms and gymnosperms occurs at the time of pollination; namely, at the time of pollination, the ovules of gymnosperms are exposed to exterior space and accessible to pollen grains, while those of angiosperms are secluded and not directly accessible to pollen grains. Although Herendeen et al. [37] tried to challenge this criterion by enumerating “unique angiosperm features” as “diagnostic features of the angiosperm crown group”, they refuted themselves by listing five angiosperms published by themselves that did not honor their own criterion [37,38]. This refutation was repeated implicitly in 2019 by Friis et al. [39,40]. Therefore, I prefer to be consistent and adopt a stricter criterion for angiosperms: angio-ovuly, namely, ovule enclosed before pollination [6,22,36]. 

It is well-known that the only foliar structures (bracts) in cones are uniformly borne on the cone axis rather than on the periphery of the cones. In contrast, the tepals in *Lingyuanthus* are arranged on the upper rim and outer surface of the cup-shaped structure (ovary/hypanthium). *Lingyuanthus* has no trace of a cone axis, which would be indispensable and required for a conifer cone. Therefore, as conifer cones are distinct from *Lingyuananthus* in morphology, it is reasonable to interpret *Lingyuananthus* as a flower.

Traditionally, Magnoliales were thought to be the most ancestral of angiosperms. Logically, apocarpy, conduplicate carpels, and follicles (features of Magnoliales) were taken as proxies of ancestrality in angiosperms [18]. In contrast to Magnoliales, flowers with an inferior ovary and hypanthium were thought to be derived or advanced in angiosperms, since these features occur only in the assumedly more derived taxa [18]. Although supported by *Archaefructus* [2,3], *Sinocarpus* [4,5], and *Neofructus* [8], with free carpels, this thinking is now challenged by *Lingyuananthus*, which is from the same formation and distinguishes itself from previously reported contemporary early angiosperms by its non-apocarpy, inferior ovary, and hypanthium. The co-occurrence of *Archaefructus* [2,3], *Sinocarpus* [4,5], *Callianthus* [14,32], *Nothodichocarpum* [10], *Sinoherba* [13], *Neofructus* [8], and now *Lingyuananthus* underscores the great diversity of flowers and angiosperms in the Yixian Formation. 

The occurrence of transmitting tissue (Figure 2b and Figure 4c) within the ovary of *Lingyuananthus* is noteworthy and intriguing. The function of transmitting tissue in living plants is guiding pollen tubes to the micropyles of ovules for successful pollination. Such function and tissue are unnecessary in gymnosperms, in which pollen grains have direct access to the micropyles. The occurrence of transmitting tissue in *Lingyuananthus* further reinforces its angiospermous affinity.

The above implications on angiosperm evolution are compatible with and reinforced by Jurassic angiosperm fossils. A feature of *Nanjinganthus* that surprised many, as seen in the title of the paper, is its “noncarpellate epigynous flower” [21]. Comparing *Lingyuananthus* and *Nanjinganthus*, it is obvious that both taxa share an inferior ovary and a hypanthium, although these features are not expected theoretically for early angiosperms [18]. It is natural for a botanist to make a difficult choice between theory and evidence, especially when they contradict each other. A wise strategy when facing such a dilemma is doing nothing and watching for further progress before taking steps. The good news liberating scientists from such a dilemma is that such awaited fossil evidence actually has been documented: independently studied and well-documented flowers, including *Monetianthus* [31]*, Divisestylus* Hermsen et al. 2003*, Antiquacupula* Sims et al. 1998*, Archaefagacea* Takahashi et al. 2008*, Calathiocarpus* Knobloch and Mai 1986*, Caryanthus* Friis 1983*, Manningia* Friis 1983*, Normanthus* Schönenberger et al. 2001, *Esgueiria* Friis et al. 1992*, Tylerianthus* Gandolfo 1998*,* and *Scandianthus* Friis and Skarby 1982 [23] from the Cretaceous, are all examples of flowers with an inferior ovary and syncarpy. All of these fossils unanimously suggest either that not all early angiosperms are apocarpous with a superior ovary, that there is another lineage of early angiosperms (namely, that angiosperms are most likely polyphyletic), that angiosperms originated much earlier, or all three. All of these implications contradict the current understanding of angiosperm evolution.

Traditionally, apocarpy, conduplicate carpels, and marginal placentation were thought to be ancestral in angiosperms. This image of ancestral angiosperms appeared to be supported by some fossil evidence, e.g., *Archaeanthus* Dilcher and Crane 1984 [41], *Archaefructus* [2,3], and *Sinocarpus* [4,5], from the Cretaceous (Table 1). However, later more careful studies indicate that *Archaeanthus* [41] and *Archaefructus* [2,3] were previously wrongly interpreted, and their support for the above image of ancestral angiosperms is spurious [42]. Numerous early angiosperms from the Yixian Formation, including *Chaoyangia* Duan [1,6]*, Callianthus* Wang and Zheng [14,32]*, Baicarpus* Han et al. 2013 [9]*, Eofructus* Han and Wang 2020 [11]*, Neofructus* Liu and Wang [8]*,* and *Liaoningfructus* Wang and Han 2011 [16] (Table 1), cast further doubt over this image. Interestingly, reviewing the history of the *Magnolia*-ancestral theory indicates that the theory was actually founded on a famous dictum of a celebrity, Goethe [43,44,45], rather than solid scientific evidence. Apparently, this theory, of great influence in the past century, needs to be updated with the progress in paleobotany and gene function studies in living plants [26,27,28,29].

Based on a wide spectrum of carpel variations in a single tree or a single flower of *Michelia* Linn., Zhang et al. proposed that a conduplicate carpel typical of Magnoliales is derived from a leaf and its axillary ovule-bearing branch [46]. This generalization was coherent with the study of *Magnolia* Linn. and *Illicium*, and was adopted by Wang [6] as one of the ways to derive carpels. However, the flowers of *Lingyuananthus*, plus *Nanjinganthus* from the Early Jurassic [21,22] and *Monetianthus*, as well as other fossil angiosperms from the Early Cretaceous [23,31], constitute hard-to-ignore exceptions to this generalization of Magnoliales. One feasible way to bypass this challenge is to derive these gynoecia in another way through floral axis invagination (also see below). 

A retrospective of botany and paleobotany indicates that Zimmermann had long applied the telome theory for the evolution of early land plants, which was successful for ferns, less so for gymnosperms, and almost of no influence for angiosperms. Its failure in angiosperms can be partially attributed to the then dominance of the *Magnolia*-ancestral theory. Now, since this roadblock has been removed by the rise of molecular systematics, it is necessary for us to reevaluate Zimmermann’s hypothesis. In his Figure 302 on page 558, Zimmermann depicted the derivation of flowers with inferior ovaries through floral axis invagination. This depiction appeared unconvincing when *Magnolia* was placed at the basalmost position in angiosperms. However, it becomes more promising now with new paleobotanical progress in mind. One of the characteristics of *Nanjinganthus* is its inferior ovary and scale-like structures on the outer surface of the ovary; such observations can be accounted for by floral axis invagination, just as Zimmermann suggested. In addition, *Monetianthus* [31], another flower independently studied, has scars of fallen-off lateral appendages on the outer surface of the ovary. Although this fossil has been well documented, the authors did not offer any explanation on the origin of its gynoecium. Parallel to *Nanjinganthus* and *Monetianthus*, the appendages on the hypanthium surface of *Lingyuananthus* reinforce such a possibility in plant evolution, although the arrangement of the lateral appendages in *Lingyuananthus* appears asymmetrical (Figure 2a and Figure 5). Wang [6], after a systematic survey of all reproductive organs in land plants, proposed several ways to derive gynoecia in angiosperms, including floral axis invagination. Now, *Lingyuananthus*, with its unique gynoecium morphology meeting the expectation of Wang [6], suggests that Zimmermann’s hypothesis appears to be a more promising hypothesis accounting for the derivation of this kind of gynoecium.

The above proposed floral axis invagination not only gives rise to this special kind of gynoecium but also introduces a special structure idiosyncratic of angiosperms, a hypanthium. This structure used to be thought to be highly derived in angiosperms, as it is remote from any structure in gymnosperms and basalmost angiosperms (*Magnolia* or *Amborella*, in traditional or current theory), and its presence in *Lingyuananthus,* in the Early Cretaceous Yixian Formation, suggests that it is not as derived as previously thought. One of the key functions of a hypanthium is to provide strong protection for the ovules/seeds within the ovary. This protection is one of many implementations of the universal trend underlying the evolution of reproduction in plants, Offspring Development Conditioning (ODC), as advanced recently by Fu et al. [35]. The derivation of conduplicate carpels, which were formerly thought to be ancestral in angiosperms, in *Michelia* (Magnoliaceae) is another good example of implementing an ODC strategy. Although appearing ubiquitous in sexually reproducing organisms, apparently, how widely applicable ODC is among other organisms is still an open question. I expect more studies to test the ODC hypothesis, using evidence of either fossil or living plants.

## 5. Conclusions

*Lingyuananthus* is a novel flower uncovered from the Yixian Formation (the Lower Cretaceous), adding to the already great diversity of angiosperms in the Yixian Formation (Lower Cretaceous). Integrated with previous reports of fossil angiosperms and recent progress made on living angiosperms, *Lingyuananthus*, with its inferior ovary, prompts botanists to reevaluate the existing hypotheses on flower derivation.

## Figures and Tables

**Figure 1 biology-11-01036-f001:**
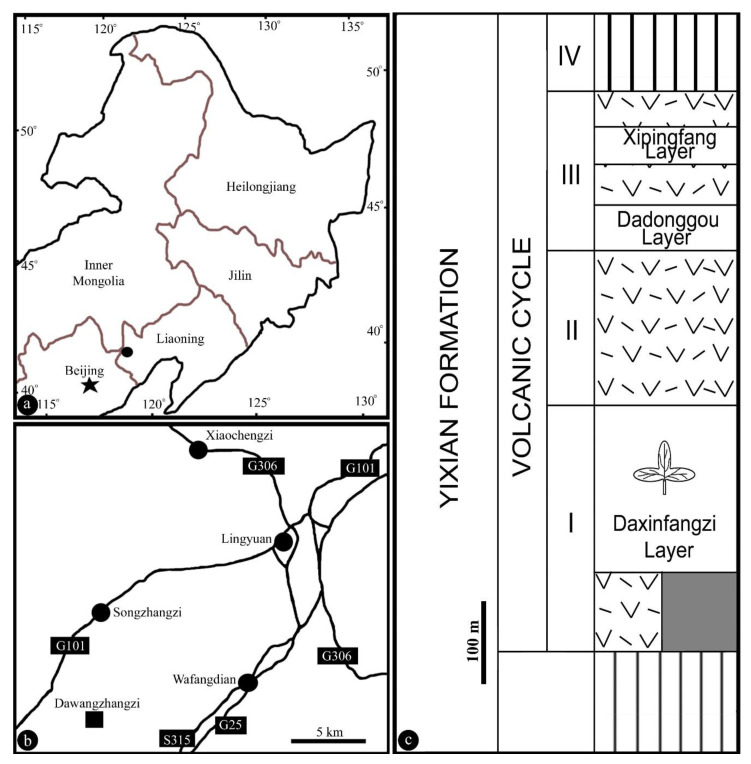
Geographical position of the fossil locality of *Lingyuananthus* gen. nov. in western Liaoning, China. (**a**,**b**), Modified from Liu and Wang (2018); (**c**), modified from Wang et al. (2021). (**a**) Fossil locality (black dot) in northeastern China; (**b**) position of fossil locality (black square) in a suburb of the city Lingyuan, Liaoning; (**c**) stratigraphic column of the Yixian Formation yielding the specimen. The column includes various volcanic eruptions and intercalating siltstones. The specimen was uncovered from the Daxinfangzi Layer.

**Figure 2 biology-11-01036-f002:**
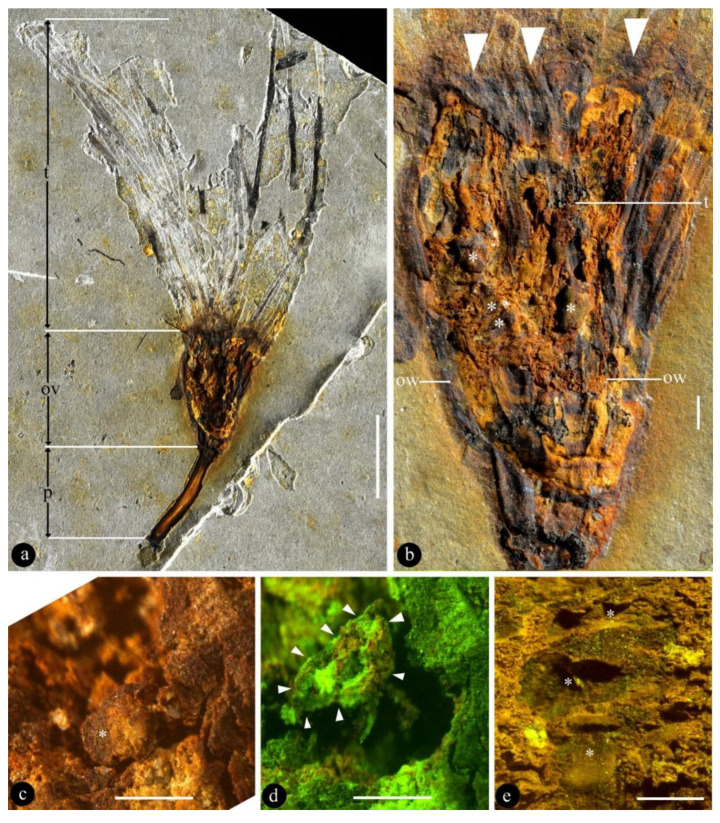
*Lingyuananthus inexpectus* gen. et sp. nov. and its details. (**a**) General view of the flower, showing pedicel (p), ovary (ov), and tepals (t). Scale bar = 1 cm. (**b**) Detailed view of the ovary, showing its obconical form, even top (triangles), ovary wall (=hypanthium) (ow), ovules (asterisks), and transmitting tissue (t). Refer to Figure 4c for details. Scale bar = 1 mm. (**c**) An ovule (asterisk) *in situ* in the ovary. Scale bar = 0.5 mm. (**d**) The ovule in Figure 2c, slightly dislocated from its original position, in the ovary. For papillae on the ovule, refer to Figure 4b. Scale bar = 0.5 mm. (**e**) Imprints left by three adjacent ovules (asterisks) of various forms in the ovary. Scale bar = 0.5 mm.

**Figure 3 biology-11-01036-f003:**
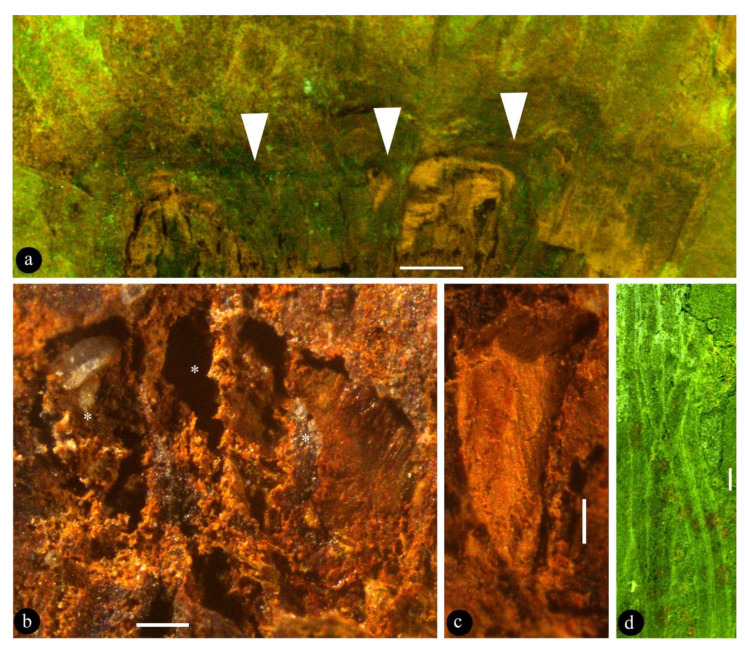
*Lingyuananthus inexpectus* gen. et sp. nov. (**a**) Flat ovary top (= floral roof, triangles) secluding the ovary. Refer to Figure 4d. Scale bar = 1 mm. (**b**) Three adjacent ovules (asterisks) of various forms in the ovary. Scale bar = 0.2 mm. (**c**) Cuneate profiled cavity left by a fallen-off ovule. Scale bar = 0.2 mm. (**d**) Several linear-shaped tepals. Scale bar = 1 mm.

**Figure 4 biology-11-01036-f004:**
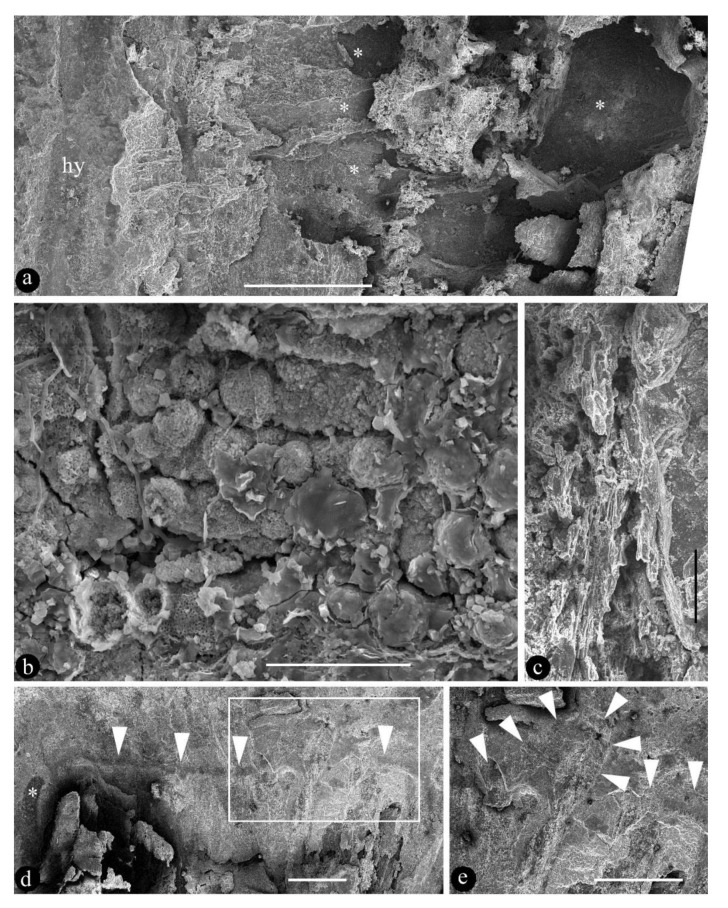
SEM views of *Lingyuananthus inexpectus* gen. et sp. nov. (**a**) A portion of the interior of the ovary showing hypanthium (hy) and cavities (asterisks) left by several ovules. Scale bar = 1 mm. (**b**) Papillae on the dislocated ovule shown in Figure 2c,d. Scale bar = 50 μm. (**c**) Transmitting tissue within the ovary. Refer to Figure 2b. Scale bar = 0.1 mm. (**d**) SEM view of the ovary top (triangles). Note one of the lateral appendages (asterisk) to the left. Refer to Figure 3a. Scale bar = 1 mm. (**e**) Detailed view of the rectangular region in Figure 4d showing the process (triangles). Scale bar = 1 mm.

**Figure 5 biology-11-01036-f005:**
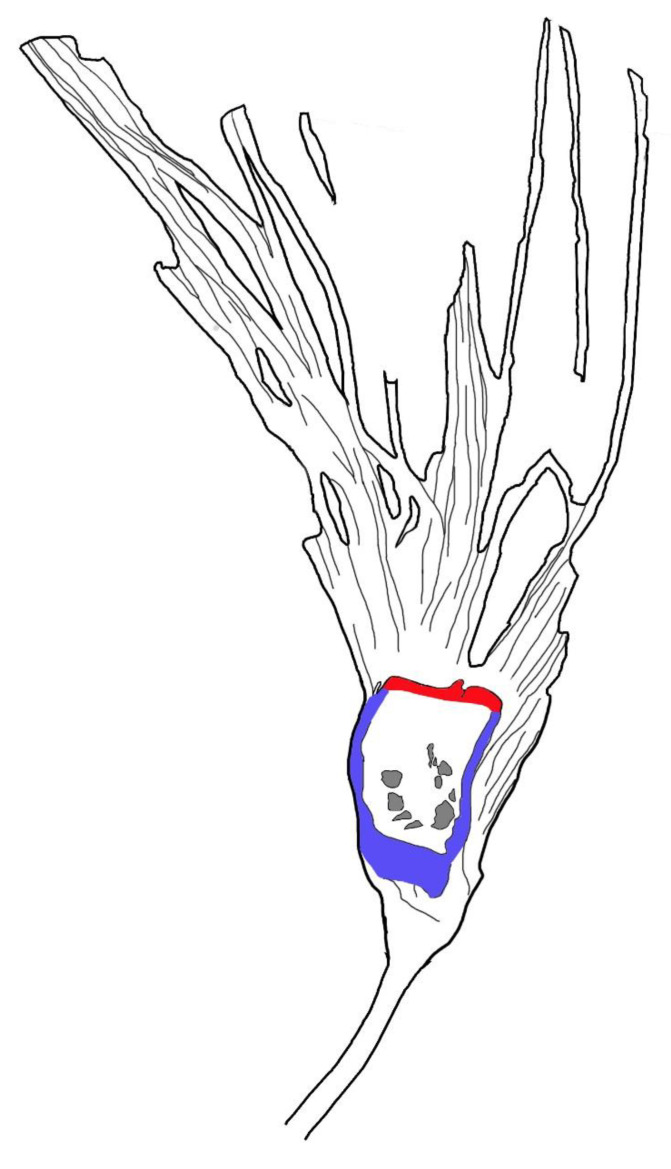
Sketch of *Lingyuananthus inexpectus* gen. et sp. nov. Note the ovule cavities (gray) in the ovary, ovary wall (=hypanthium) (blue), and floral roof (red) secluding the ovary.

**Table 1 biology-11-01036-t001:** Comparison between *Lingyuananthus inexpectus* gen. et sp. nov. and other angiosperms from the Yixian Formation.

	*Archae-fructus sinensis*	*Sinocarpus decussatus*	*Nothodicho-carpum lingyuanensis*	*Neofructus lingyuanensis*	*Callianthus dilae*	*Baicarpus*	*Sinoherba ningchengen-sis*	*Lingyuananthus inexpectus* gen. et sp. nov.
Sexuality	Bisexual	Female	Bisexual	Female	Bisexual	Female	Female	Female
Ovule enclosed	Yes	Yes	Yes	Yes	Yes	Yes	Yes	Yes
Carpellate	Yes	Yes	Yes	Yes	?	Yes	Yes	No
Carpel arrangement	Free	Semi-free	Free	Free	?	Free	?	No
Carpel number	Multiple	4	2	Multiple	2	3+	?	?
Reference	Sun et al., 1998, 2002	Leng and Friis, 2003	Han et al., 2017	Liu and Wang, 2018	Wang et al., 2021	Han et al., 2013	Liu et al., 2021	present study

## Data Availability

The specimen is deposited in the Herbarium, Nanjing Institute of Geology and Palaeontology, Nanjing, China (http://www.nigpas.cas.cn/).

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
