# Peer review of "A Novel Early Cretaceous Flower and Its Implications on Flower Derivation"

_biology, 2022, doi:10.3390/biology11071036_

Round 1
Reviewer 1 Report
-The figure 1 is not cited in the text. A figure illustrating the stratigraphic origin of the fossil was added; however, it remains unexplained in the text. A caption presenting the diverse types of lithology illustrated in Fig 1C is missing. Scale bar missing in the stratigraphic column (fig 1C).
-In the systematic section, it looks curious to have two “remarks” sections (one after the description of the genus, the other after the description of the species). I suggest to group all these informations in a single section after the species description.
Author Response
Thanks.
-The figure 1 is not cited in the text. A figure illustrating the stratigraphic origin of the fossil was added; however, it remains unexplained in the text. A caption presenting the diverse types of lithology illustrated in Fig 1C is missing. Scale bar missing in the stratigraphic column (fig 1C).
Figure 1 has been cited. A brief explantion for Fig. 1c has been added. A scale has been added in the figure.
-In the systematic section, it looks curious to have two “remarks” sections (one after the description of the genus, the other after the description of the species). I suggest to group all these informations in a single section after the species description.
These two remarks have been merged, following this suggestion.
Reviewer 2 Report
The author has modified the manuscript, and the new version is much improved. Added figures convinced me that this fossil flower has a hypanthium. However, there are important inaccuracies in the text and figure captions that should be corrected.
Sepals, petals, stamens, pistils departs from the flower receptacle (not from the ovary). Hypanthium is a special structure of the lower parts of the integument and androecium, which is formed in some plants as a result of the expansion of the receptacle and the fusion of the flower tube with it (the fused bases of the tepals and stamens). The fact that the petals of Lingyuananthus gen. et sp. nov. arise directly from the sheath that surrounds the cavity with the ovules proves that this sheath is a hypanthium. In this regard, the following should be corrected in the manuscript:
- The petals are not located along the outer edge of the ovary, but arise from the edge of the hypanthium (lines 75, 94, 98, 104, 151, 152, 155, 222 needs to be corrected).
- The ovary and hypanthium are different structures. Therefore, it is not correct to write "ovary wall (=hypanthium)". According to author’s figures the hypanthium wall is observed on the presented samples, since the petals extend from it. And the cavity containing the ovules is the ovary. In this regard, designation “ow” should be replaced by hypanthium in figure captions.
After correcting these comments, this manuscript can be accepted for publication in Biology.
Author Response
Thanks for your help.
The author has modified the manuscript, and the new version is much improved. Added figures convinced me that this fossil flower has a hypanthium. However, there are important inaccuracies in the text and figure captions that should be corrected.
Sepals, petals, stamens, pistils departs from the flower receptacle (not from the ovary). Hypanthium is a special structure of the lower parts of the integument and androecium, which is formed in some plants as a result of the expansion of the receptacle and the fusion of the flower tube with it (the fused bases of the tepals and stamens). The fact that the petals of Lingyuananthus gen. et sp. nov. arise directly from the sheath that surrounds the cavity with the ovules proves that this sheath is a hypanthium. In this regard, the following should be corrected in the manuscript:
1. The petals are not located along the outer edge of the ovary, but arise from the edge of the hypanthium (lines 75, 94, 98, 104, 151, 152, 155, 222 needs to be corrected).
All these terms have been changed according to this suggestion.
2. The ovary and hypanthium are different structures. Therefore, it is not correct to write "ovary wall (=hypanthium)". According to author’s figures the hypanthium wall is observed on the presented samples, since the petals extend from it. And the cavity containing the ovules is the ovary. In this regard, designation “ow” should be replaced by hypanthium in figure captions.
This designation has been changed.
After correcting these comments, this manuscript can be accepted for publication in Biology.
Reviewer 3 Report
Undoubtedly, the manuscript greatly improved. But I have two serious remarks about the manuscript.
The first, regarding to point 4 of my review on the article. The article is devoted to a fossil plant. And if sepals are not visible on the fossil, this does not prove that they were not on a living plant, as well as that they were. Petals are clearly visible on the sample under study. However, the author cannot prove whether the perianth was differentiated or not. But in diagnosis author noticed that the "flower incomplete" (line 75). This frase is a very categorical and not justified statement. Therefore, I think that in this case it is better not to talk about the differentiation of the perianth at all and to use the term "petals" instead of "tepals" (a term that combines sepals and petals).
My second remark concerns the author's point of view that "ovary wall = hypanthium" (lines 81, 139, 178, 237). This is a wrong opinion, because these are different structures - the ovary is enclosed within the hypanthium. From this follows another erroneous assertion of the author that the "tepals on ovary rim" (line 77). On the investigated fossil we are clearly seen only petals. Petals cannot locate on the ovary rim – they are on the rim of hypanthium. The author needs to correct the wordings in the manuscript according to these remarks.
In the whole, I believe that the manuscript warrants publication in Biology.
Author Response
Thanks for your help.
Undoubtedly, the manuscript greatly improved. But I have two serious remarks about the manuscript.
The first, regarding to point 4 of my review on the article. The article is devoted to a fossil plant. And if sepals are not visible on the fossil, this does not prove that they were not on a living plant, as well as that they were. Petals are clearly visible on the sample under study. However, the author cannot prove whether the perianth was differentiated or not. But in diagnosis author noticed that the "flower incomplete" (line 75). This suggestion has been foolowed. This frase is a very categorical and not justified statement. Therefore, I think that in this case it is better not to talk about the differentiation of the perianth at all and to use the term "petals" instead of "tepals" (a term that combines sepals and petals).
I have changed all terms as "tepals".
My second remark concerns the author's point of view that "ovary wall = hypanthium" (lines 81, 139, 178, 237). This is a wrong opinion, because these are different structures - the ovary is enclosed within the hypanthium. From this follows another erroneous assertion of the author that the "tepals on ovary rim" (line 77). On the investigated fossil we are clearly seen only petals. Petals cannot locate on the ovary rim – they are on the rim of hypanthium. The author needs to correct the wordings in the manuscript according to these remarks.
This error has been fixed.
In the whole, I believe that the manuscript warrants publication in Biology.
Reviewer 4 Report
The author presents a supposed new genus and species of angiosperms from the Yixian formation, and most of the manuscript is dedicated to a discussion of the implication of the inferred morphology on the evolution of carpels. However, there are fundamental issues with the study that need to be addressed before any biological interpretation might be advanced.
The angiospermous nature of the fossil is far from being established. The structures that the authors claim to be ovules, if they are structures at all, are all of differing shape and orientation. The post-hoc justification from the author that this is due to different “developmental stages” is far from convincing, and not supported by the data themselves. The linear “tepals”, that the author claim to be hypanthial, seem to attach to the base of the structure in figure 1a and 1b.
The problem with the interpretation of the fossil is further complicated by its uniqueness (only a single specimen is shown) and its preservation status. The abundant reddish coloration suggests some sort of iron incrustation, which makes the interpretation of the structures shown as biological even more complicated. The papillae shown in figure 4b might be pyrite framboids or similar structures. An EDX analysis is needed to further confirm that the structures observed are indeed carbon, and not taphonomic. Further specimens of the same structures might also be needed before properly erecting a new genus and species.
Author Response
Thanks for your help.
The author presents a supposed new genus and species of angiosperms from the Yixian formation, and most of the manuscript is dedicated to a discussion of the implication of the inferred morphology on the evolution of carpels. However, there are fundamental issues with the study that need to be addressed before any biological interpretation might be advanced.
The angiospermous nature of the fossil is far from being established. The structures that the authors claim to be ovules, if they are structures at all, are all of differing shape and orientation. The post-hoc justification from the author that this is due to different “developmental stages” is far from convincing, and not supported by the data themselves. I have no other choice but to use the term "ovule". I wonder what would be the proper term if the reivewer were me. The linear “tepals”, that the author claim to be hypanthial, seem to attach to the base of the structure in figure 1a and 1b.
My previous description is not accurate. I have fixed this problem.
The problem with the interpretation of the fossil is further complicated by its uniqueness (only a single specimen is shown) and its preservation status. This is a routine in palaeobotany, as seen in publications on Science and Nature. The abundant reddish coloration suggests some sort of iron incrustation, which makes the interpretation of the structures shown as biological even more complicated. The papillae shown in figure 4b might be pyrite framboids or similar structures. An EDX analysis is needed to further confirm that the structures observed are indeed carbon, and not taphonomic. Such papilae are strictly restricted to portion of the ovule. It is unlikely to be pyrite framboids, which should be universal rather than restrcited to ovules in the plant organ. Further specimens of the same structures might also be needed before properly erecting a new genus and species. I do welcome more studies of more specimens. Please remember that, if this paper were not the first documentation, how could more specimens be published as ensuing materials (not the first materials)? Let us be the first one, and expect the second and more to come.
This manuscript is a resubmission of an earlier submission. The following is a list of the peer review reports and author responses from that submission.
Round 1
Reviewer 1 Report
This is an interesting paper presenting a new species of flower from the Barremian-Aptian of China.
My main concerns are:
-The lack of a geological setting section.
-Some points of the description that could be more detailed.
-Some contradictions between the conclusions of the paper and the results.
-The quality of figures.
Please see details, bellow:
-Lines 60, 64 and 67: « in the Formation» >>> « formation » not in capital letter.
-MATERIAL AND METHODS: A section describing the geological, lithological and palaeoenvironmental context of the Yixian Formation is missing. I think such information are capital for the international readership.
-DESCRIPTION: Can you give information about the gross morphology of the flower ? I suggest to describe the structures from the outermost to the innermost (1) Perianth (2) ovary. Line 102: any information about the number of whorls of tepals ? The fig 1A being very blurry is not possible to clearly identify pieces of the perianth on the photograph. Line 104: ovule “various forms” >>> specify ; since the pictures are blurry in figure 1 and the fossil quite poorly preserved, it is not easy to be convinced by the nature of the described structures. Line 105: what is the shape, the organization and the size of papillae ? Not clear on fig1F.
-Line 97: “coaly compression embedded in yellowish siltstone” = repetition with the Material & Methods section.
-Line 98: “The fossil is 6.4 cm long, 3.8 cm wide” = repetition with the Material & Methods section.
-Line 122: add a reference ?
-Line 142: you wrote “the tepals in Lingyuanthus are differentiated into three types”…I don’t understand, in the description you wrote “not differentiated in size and morphology”. It is not concordant.
-Line 246: you wrote “Lingyuanfructus is novel fruit type”…I don’t understand, you wrote line 112 “This leads us to interpret the fossil as a flower rather than a fruit”. Here the conclusion is not consistent with the text.
-Figure 1: in the pdf version of the manuscript the figures 1A to 1G are very blurry. I recommend better quality of images…without that it is very difficult clearly identify the structures described in the text. Specify the kind of microscopy used in fig.1f: SEM I suppose ? Since the figure 1F to 1G are totally blurry it is not possible to distinguish ovules or papillae described in the caption.
-Figure 2: scale bars missing
Reviewer 2 Report
Comments to the manuscript
An Early Cretaceous Novel Flower Suggests a Unique Way to Derive Flowers
Dr. Xin Wang desceribed their find of angiosperm flower with hypanthium, inferior ovary, and ovules inside its ovary from the Yixian Formation (China). This is an important discovery, in term of the origin and evolution of angiosperms' flowers, since the mainstream thinking is thatthat the diversity of flowers comes from the Magnoliales flower. The study has great potential to be published and it should be, but for the data to be valid and robust, several parts of the publication need to be improved. My recommendations would be the following.
- The presence of hypanthium in this find is questionable. It is not clear in Figures 1a and 1b which structures are interpreted as hypanthium. The dark border observed in Figure 1b can be both the wall of the hypanium and the wall of the ovary. It is necessary to mark in Figure 1 the structures that you interpret as hypanthium.
- For a clearer understanding of your sample, it is necessary to give a schematic drawing of a flower indicating the structures observed on the impression.
- It is necessary to add an overview map depicting the location of this find and give a geological column with reference to samples to a specific layer or give a reference, if this information was published earlier.
- It is necessary to show by arrows in fig. 1b position of ovules.
- Figure 2 doesn't make much sense in conclusion. I suggest removing it.
- The text of the manuscript is overly emotional. I advise you to soften some of the wording (listed below).
- Lines 20–21: “New fossil material with its unique 20 character assemblage challenges the currently wide-accepted theories of angiosperm evolution”.
Try to express your thoughts more gently.
- Lines 33–34: “Apparently Sokoloff et al. have confused them with angiosperms and eudicots, which are two concepts distinct for any botanical students”.
Try to express your thoughts more gently.
- Lines 39–40: “Their conclusion is apparent a consequence of carelessness in observation…”
The phrase contains a negative emotional connotation. Please rephrase.
- Lines 104–105: “Multiple (more than three) ovules of various forms are em-104 bedded in the ovary…”
These ovules belong to the same species and should be of the same shape. Perhaps the difference in the shape of the ovules is caused by a different degree of preservation? Please clarify your point of view.
- Lines 109–110: “The inner bodies are millimetric in size, more or less round-triangular…”
The description of the taxon on lines 104–105 indicates that the ovules have a various forms. I propose to clarify the shape of the ovules in the Description.
- Line 110: “…so they are interpreted as ovules because any trace of seed coat…”
It is necessary to show traces of the seed coat in Figure 1 and add this information to the Description.
- Lines 132–133: “Therefore distinguishing angiosperms and gymnosperms using angiospermy alone is not safe.”
What do you mean? Rephrase this sentence.
- Lines 142–144: “In contrast, the tepals in Lingyuanthus are differentiated into three types and are arranged on the upper margin of the bowl-shaped structure.”
However, the description of the taxon indicates that the tepals are of the same type (lines 102–103). Make it clear.
- Lines 197–199: “Apparently, this old-fashioned gratuitous theory of great influence in the past century should be discarded, especially when with the great progress in palaeobotany and gene function studies in mind.”
The phrase contains a negative emotional connotation. Please rephrase.
I do not agree with you that this theory can be discarded as having lost its meaning for Botany. It is likely that the origin of Lingyuanthus can be explained by Zimmermann's theory. However, this does not prove that the flowers of all modern Rosidae originated according to this theory alone.
What did you mean by researching the functions of genes in botany? A reference is required.
- Lines 213–214: “Now since such a road block has been removed by the rising molecular systematics, it is not harmful for us to re-evaluate Zimmermann’s hypothesis.”
“it is not harmful” should be “it is necessary for us”.
- Lines 216–217: “Such a depicting appeared ridiculous when Magnolia was placed at the 216 basalmost position in angiosperms.”
“appeared ridiculous” should be “looks unconvincing”.
- Lines 225–226: “Now 225 Lingyuananthus with its unique gynoecium morphology is shouting out…”
An overly emotional proposition. Please rephrase.
- Lines 225–227: “Now Lingyuananthus with its unique gynoecium morphology is shouting out: Zimmermann’s hypothesis may be the only right way to derive such a kind of gynoecium.”
This statement is too categorical. It is possible that Lingyuananthus originated according to Zimmermann's theory, but flowers of this type in modern Angiosperms could have been derived from flowers of the Magnoliidae in another alternative way (see comment 15).
General review. I find the data interesting and encourage the author to make the necessary changes to publish it. I hope my comments are helpful to you.
Reviewer 3 Report
The article can be published after correction and addition in according to the comments.
Review on the article
Xin Wang
“An Early Cretaceous Novel Flower Suggests a Unique Way to Derive Flowers”
Recently, articles on fossil angiosperms of the Pre-Cretaceous and Early Cretaceous have become more frequent. Perhaps fossil material with unique set of features have been found earlier but these finds, probably, were simply ignored, since this fossil material was out of the accepted theories about the origin and evolution of angiosperms. I share the point of view that angiosperms could have originated in the Jurassic and even earlier. However, there are a lot of scientists who disagree with this point of view. Therefore, I agree with the author again that research on fossil angiosperms requires a very honest and careful attitude, and published materials should be as clear and convincing/evidence-based as possible.
In general, I have a very good impression about this article, but I have several serious comments to that. First, I recommend using less categorical expressions regarding the material studied, because conclusions based on single sample only look a bit speculative.
The finding of Lingyuananthus inexpectus gen. et sp. nov. is truly unique and important for the formation of a new understanding at the history of the origin of angiosperms. Therefore, an article about this fossil should contain graphic material (maps, schemes) and illustrations to better understand what the author exactly saw, studying the sample. It is very difficult to identify all structures on the fossil, having the photographs only.
- On the lines 18, 66, 156 and further in the text the author writes about the presence of hypanthium in Lingyuananthus inexpectus gen. and sp. nov., but there are no reliable evidences of the presence of hypanthium on the fossil in the text, and this structure do not indicated on the fig. 1a-b.
- lines 29, 55, 74–76, 152–153, … In my opinion, at each first mention of a taxon (genus or species), it is necessary to indicate author and the year of its description.
- line 63. It is necessary to provide references about the age of Yixian Formation. Also, it would be useful to provide an illustration with the geographical location of the Yixian Formation, and a draw of the geological composition of the Formation with the layer from which the sample was taken indicated.
- lines 65. What does the author mean, writing “…numerous linear tepals…”? Does it mean sepals with petals or only petals? On fig. 1a-b I see pedicle, receptacle, ovary, petals, but do not see sepals. Were they missing?
- lines 82–84, 96–106. The diagnosis should be more detailed, using generally accepted botanical terms. The type of flower (actinomorphic, zygomorphic or asymmetric) should be indicated. Also the author does not write anything about the receptacle, which can be seen in Fig. 1a-b. If it is possible, the type of perianth: double/simple (simple corolla or simple calyx) should be indicated. It is necessary to add information about sepals and hypanthium to the description. In my opinion, judging by the photo, the petals were not numerous (see line 83). The structures mentioned in the description are not indicated on the Fig. 1. For better clarity and understanding of the description, it is necessary to give at least a schematic reconstruction (draw) of the taxon described with the designation of the structures that the author observed on the fossil. Figure of ovules (2a-c) are not informative and should be removed from the article. The diagnosis should include a comparison of the Lingyuananthus inexpectus and sp. nov. with some other known fossil angiosperms.
- lines 157–159. "Unlike the follicle-like fruits seen in Archaefructus, Sinocarpus, and Neofructus, the fruit derived from the ovary of Lingyuananthus would be a pepo, which used to be out of the list of features idiosycratic of ancestral angiosperms” – is very categorical and not justified statement.
- line 246. The article is mainly devoted to a new taxon from the Early Cretaceous – Lingyuananthus inexpectus and sp. nov. However, in Сonclusions, the author writes about any other taxon (Lingyuanfructus), and does not even mention Lingyuananthus. At the same time, there are no mentions about the genus Lingyuanfructus in the Abstract.
Reviewer 4 Report
While researching the cited literature, it became clear to me that the topic of fossil angiosperm is highly contested. I was not aware of that. Amidst refusals and rebuttals, one author laments "How can such obvious errors be made and escape the attention of the reviewers? This is a question deserving attention of everyone." I will do my best to raise to the task and be as critical and diligent as possible.
- The presented fossil is likely not a complete flower. This is not addressed. The ovules are clearly visible in the carpels, and the carpels in profile show a considerable thickness. Such thick walls do not become translucent during drying, and mostly likely not during fossilisation. So I have to assume that part of the lower, darker portion of the fossil is missing, probably stuck to the counter-plate of the rock.
- The fact that the carpels are so obvious leads me to conclude that a calyx or hypanthium, if truly originally part of the flower, is also missing in that parts and can only be seen in the underlying layers, visible through the tepals or along the margins of the receptacle. The dark shade suggests that but no detail of that area is provided.
- The upper margin of the fused ovary in Fig. 1b looks like an artifact, a discoloration. I concede that there seem to be two paler triangular structures that can be interpreted as the tips of sepal (calyx). But it could also be a long broad stigma as in the carpels of an Austrobaileyaceae. Overall the flower reminds me strongly of Xylopia sp. as dried specimen. The true fusion of the calyx/hypanthium cannot be discerned from the prictures provided; the mere presence of a closed hypanthium may be even doubted.
- In the introduction, "numerous" petals are announced but later only few are confirmed in the picture. I think they can be counted. The veination of the petals/tepals are not described.
- A better illustration of the fossil opposite a diagram of the interpreted structures would be necessary to convince people. The sketches of the separate ovules do not help much (Fig. 2)
Most of the introduction and discussion is about the presence or absence of trustworthy angiosperm fossils from the Jurassic. This has little to nothing to do with the present fossil, which is admittedly from the Early Cretaceous. So, the fossil is definitely an angiosperm, but it does not fill the gap. Whether or not the earlier fossils from the Jurassic are angiosperms is a moot point when this fossil here is concerned.
The next issue is whether or not the carpels are fused to an inferior ovary in this fossil. As I said above, I cannot follow this interpretation whithout doubt. But again this does not overthrow the predominant theory of free carpels as ancestral state in angiosperms as most parsimonous explanation. We have examples of fused inferior ovaries in such an early diverging lineage as the Nymphaeaceae; evolution can go the same way many times independently. The diversity of the Early Cretaceous angiosperms that are accepted by both sides of the argument show a morphological variation that necessitates a certain evolutionary fore-run extending into the upper Jurassic at least. This is indicated by most time calibrated phylogenies.
The cited literature about the true nature of the angiosperm gynoceum (sporophyll vs. axis) was interesting and enlightening. However, the presented data seemed trustworthy only for extant taxa.
The authors states that there is a mantra of "no angiosperms before the Cretaceous" but actually the authors of these cited papers are mostly merely stating that confidentially assignable angiosperms, which show all flower components clearly and may be aligned with one or the other extant family, are so far only found in Cretaceous rocks. I guess these authors would be more than happy to accept a Jurassic angiosperm when it is convincinly presented. Unfortunately, the presentation of observed object and interpeted reconstruction often involves a huge leap of faith.
The style and wording of the text appear to be of novice level, which surprising as the author has an ample publication record including several pricey books. I resent the use of adverbs like "ironically", "surprisingly" and the like; this should not be used in a natural scientific text, at least not as often and in the context as done here. Usage of the gerund of a verb instead of the noun, missing articles (definite and indefinite), and some omitted/superfluous words that change the meaning of the sentence are the most frequent issues that I found (marked in the text).
